# Towards Efficient Communication and Secure Federated Recommendation System via Low-rank Training

## ABSTRACT

With increasing regulatory constraints on centralized data gathering, federated recommendation (FedRec) systems have emerged as a promising solution for safeguarding user privacy. However, the deployment of FedRec introduces challenges in communication efficiency, stemming from the need to transmit neural network models between individual user devices and a central server. This study addresses prior shortcomings in efforts to enhance communication efficiency in FedRec. Common approaches often lead to issues such as computational overheads, inadvertent disclosure of sensitive user information, model specificity constraints, and compatibility issues with secure aggregation protocols. In response, we propose a novel framework inspired by Parameter-Efficient Fine-tuning. Our framework leverages the concept of adjusting lightweight trainable parameters while keeping most pre-trained parameters frozen. This innovative approach yields a substantial reduction in both uplink and downlink communication overheads while avoiding the introduction of additional computational burdens. Critically, our framework remains fully compatible with secure aggregation protocols, including the robust use of Homomorphic Encryption. This research offers a promising avenue to address the drastic need for efficient and secure FedRec systems, ensuring user privacy protection in an era of stringent data regulations. Extensive experimental results and analysis on multiple datasets across various model architectures and security mechanisms validate the effectiveness of the proposed method.

## CCS CONCEPTS

• **Do Not Use This Code → Generate the Correct Terms for Your Paper**; *Generate the Correct Terms for Your Paper*; Generate the Correct Terms for Your Paper; Generate the Correct Terms for Your Paper.

## KEYWORDS

Recommendation System, Federated Learning, Communication efficiency

**ACM Reference Format:**
Anonymous Author(s). 2023. Towards Efficient Communication and Secure Federated Recommendation System via Low-rank Training. In *Proceedings of ACM Conference (Conference'17)*. ACM, New York, NY, USA, 12 pages. https://doi.org/10.1145/nnnnnnn.nnnnnnn

## 1 INTRODUCTION

In a centralized recommendation system, all user behavior data is collected on a central server for training. However, this method can potentially expose private information that users may be hesitant to share with others. As a result, various regulations such as the General Data Protection Regulation (GDPR)[22] and the California Consumer Privacy Act (CCPA)[18] have been implemented to limit the centralized collection of users' personal data. In response to this challenge, and in light of the increasing prevalence of edge devices, federated recommendation (FedRec) systems have gained significant attention for their ability to uphold user privacy.

The training of FedRec involves numerous edge devices, such as mobile phones, laptops, and PCs. This scenario is often referred to as cross-device federated learning (FL). However, training large models on edge devices poses challenges due to unreliable connections and the need to transfer a recommendation model between each user's device and a central server. This task is becoming increasingly challenging due to increasing model complexity and parameters in modern recommendation systems [17]. In addition, clients participating in federated systems often exhibit differences in their computational processing speeds and communication bandwidth capabilities, primarily stemming from variations in their hardware and infrastructure. These discrepancies can give rise to stragglers and decrease the number of participants involved in training, potentially leading to diminished system performance.

Practical FedRec systems require implementing mechanisms that decrease the amount of communication relative to computation. Three commonly employed approaches for reducing communication costs include (i) diminishing the frequency of communication by permitting local updates, (ii) minimizing message size through message compression, (iii) reducing server-side communication traffic by restricting the number of participating clients per round [23]. Importantly, these three methods are independent and can be combined for enhanced efficiency.

In this study, we address the challenge of communication efficiency in federated recommendations by introducing an alternative to compression methods. While compression techniques aim to reduce the volume of data transmitted between clients and servers, however, many existing compression methods involve encoding and decoding steps that can introduce significant delays, potentially outweighing the gains achieved in per-bit communication time [21]. Another crucial consideration is the compatibility with aggregation protocols. For example, compression techniques that do not align with the all-reduce approach may yield reduced communication efficiency in systems employing all-reduce style aggregation techniques [21]. This is also necessary for many secure aggregation protocols such as Homomorphic Encryption (HE) [4]. Moreover, many algorithms assume clients have the same computational power, but this may induce stragglers due to computational heterogeneity and can increase the runtime of algorithms.

Based on our observation that the update transferred between clients and a central server has a low-rank structure bias, we propose Correlated Low-rank Structure update (CoLR). CoLRincreases communication efficiency by adjusting lightweight trainable parameters while keeping most pre-trained parameters frozen. Under this training scheme, only a small amount of trainable parameters will be shared between the server and clients and can greatly reduce the amount of communication needed to transmit a model update. Compared with other compression techniques, our methods have the following benefits. (i) **Reduce both up-link and down-link communication cost:** CoLR avoid the need of unrolling the low-rank message at the aggregation step by using correlated projection to reduce server runtime and downlink message size, (ii) **Low computational overheads:** Our method enforces a low-rank structure on the local update during the local optimization stage so it removes the need to perform an expensive decomposition step. Moreover, CoLR can be integrated into common aggregation methods such as FEDAVG and does not require additional computation. (iii) **Compatible with secure aggregation protocols:** the aggregation step on CoLR can be carried by simple additive operations, this simplicity makes it compatible with strong secure aggregation methods such as HE, (iv) **Bandwidth Heterogeneity Awareness:** Allowing adaptive rank for clients based on computational and communication budget.

Our contributions can be summarized as below:

- We propose a novel framework, CoLR, designed to tackle the communication challenge in training FedRec systems.
- Our framework demonstrates a capability to provide a strong foundation for building a secure and practical recommendation system. Specifically, CoLR is compatible with robust secure aggregation protocols and hence, reinforces the security of the overall recommendation systems.
- We conducted experiments to showcase the effectiveness of CoLR. Notably, even with an update size equates to 6.25% of the baseline model, CoLRdemonstrates remarkable efficiency by retaining 93.65% accuracy (in terms of HR) compared to the much larger baseline.

## 2 RELATED WORK

*Federated Recommendation (FedRec) Systems.* In recent years, FedRec systems have risen to prominence as a key area of research in both machine learning and recommendation systems. FCF [2] and FedRec [14] are the pioneering FL-based methods for collaborative filtering based on matrix factorization. The former is designed for implicit feedback, while the latter is for explicit feedback. To enhance user privacy, FedMF [5] applies distributed matrix factorization within the FL framework and introduces the HE technique for securing gradients before they are transmitted to the server. MetaMF [15] is a distributed matrix factorization framework using a meta-network to generate rating prediction models and private item embedding. [25] presents FedPerGNN, where each user maintains a GNN model to incorporate high-order user-item information. FedNCF [19] adapts Neural Collaborative Filtering (NCF) [9] to the federated setting, incorporating neural networks to learn user-item interaction functions and thus enhancing the model's learning capabilities.

*Communication Efficient Federated Recommendation.* Communication efficiency is of the utmost importance in FL [11]. Some works explore reducing the entire item latent matrix payload by meta-learning techniques [15, 24]. For example, MetaMF [15] adopts the meta recommender to deploy smaller models on the client to reduce memory consumption. LightFR [26] proposes a framework to reduce communication costs by exploiting the learning-to-hash technique under federated settings and enjoys both fast online inference and economic memory consumption. However, these

*Low-rank Structured Update.* Konečný et al. [11] propose to enforce every update to local model $\Delta_u$ to have a low rank structure by express $\Delta_u = A_u^{(t)} B_u^{(t)}$ where $A_u^{(t)} \in \mathbb{R}^{d_1 \times k}$ and $B_u^{(t)} \in \mathbb{R}^{k \times d_2}$. In subsequent computation, $A_u^{(t)} \in \mathbb{R}^{d \times k}$ is generated randomly and frozen during a local training procedure. In each round, the method generates the matrix $B_u^{(t)}$ afresh for each client independently. This approach saves a factor of $d_1/k$. They interpret $B_u^{(t)}$ as a projection matrix, and $A_u^{(t)}$ as a reconstruction matrix. Hyeon-Woo et al. [10] proposes a method that re-parameterizes weight parameters of layers using low-rank weights followed by the Hadamard product.

$$W = W_1 \odot W_2 = \left( X_1 Y_1^\top \right) \odot \left( X_2 Y_2^\top \right)$$

Given this parameterization, the rank of $W$ is upper bound by $\text{rank}(W_1)\,\text{rank}(W_2)$ is less constrained than a conventional low-rank parameterization $W = XY^\top$. The authors show that FedPara can achieve comparable performance to the original model with 3 to 10 times lower communication costs on various tasks, such as image classification, and natural language processing.

*Secure FedRec.* Sending updates directly to the server without implementing privacy-preserving mechanisms can lead to security vulnerabilities. Chai et al. [5] demonstrated that in the case of the Matrix Factorization (MF) model using the FedAvg learning algorithm, if adversaries gain access to a user's gradients in two consecutive steps, they can deduce the user's rating information. Therefore, it is crucial to incorporate privacy-preserving mechanisms for the update parameters transmitted from clients to the server. One approach, as proposed by Chai et al. [5], involves leveraging HE to encrypt intermediate parameters before transmitting them to the server. This method effectively safeguards user ratings while maintaining recommendation accuracy. However, it introduces significant computational overhead, including encryption and decryption steps on the client side, as well as aggregation on the server side, which is performed on ciphertext. Approximately 95% of the time consumed by Chai et al. [5] system is dedicated to server updates, where all computations are carried out on the ciphertext.

## 3 PRELIMINARIES

In this section, we present the preliminaries and the setting that the paper is working with. Also, this part will discuss the challenges in applying compression methods.

### 3.1 Federated Learning for Recommendation

In the typical settings of item-based FedRec systems [14], there are $M$ users and $N$ items where each user $u$ has a private interaction set denoted as $O_u = \{(i, r_{iu})\} \subset [N] \times \mathbb{R}$. These users want to jointly

build a recommendation system based on local computations without violating participants' privacy. This scenario naturally aligns with the horizontal federated setting [16], as it allows us to treat each user as an active participant. In this work, we also use the terms user and client interchangeably since each user is equivalent to one client. The primary goal of such a system is to generate a ranked list of top-K items that a given user has not interacted with and are relevant to the user's preferences. Mathematically, we can formalize the problem as finding a global model parameterized by $\theta$ that minimizes the following global loss function $\mathcal{L}(\cdot)$:

$$\mathcal{L}(\theta) \triangleq \sum_{u=1}^{M} w_u \mathcal{L}_u(\boldsymbol{\theta}) \tag{1}$$

where $\theta$ is the global parameter, $w_u$ is the relative weight of user $u$. And $\mathcal{L}_u(\theta) := \sum_{i \in O_u} \ell_u(\theta, (i, r_{ui}))$ is the local loss function at user $u$'s device. Here $(i, r_{ui})$ represents a data sample from the user's private dataset, and $\ell_u$ is the (non-convex) loss function defined by the learning algorithm. Setting $w_u = N_u/N$ where $N_u = |O_u|$ and $N = \sum_{u=1}^{M} N_u$ makes the objective function $\mathcal{L}(\theta)$ equivalent to the empirical risk minimization objective function of the union of all the users' dataset. Once the global model is learned, it can be used for user prediction tasks.

In terms of learning algorithms, Federated Averaging (FedAvg) [16] is one of the most popular algorithms in FL. FedAvg divides the training process into rounds. At the beginning of the $t$-th round ($t \geq 0$), the server broadcasts the current global model $\theta^{(t)}$ to a subset of users $\mathcal{S}^{(t)}$ which is often uniformly sampled without replacement in simulation [14, 23]. Then each sampled client in the round's cohort performs $\tau_u$ local SGD updates on its own local dataset and sends the local model changes $\Delta_u^{(t)} = \theta_u^{(t, \tau_u)} - \theta^{(t)}$ to the server. Finally, the server performs an aggregation step to update the global model:

$$\theta^{(t+1)} = \theta^{(t)} + \frac{\sum_{u \in \mathcal{S}^{(t)}} p_u \Delta_u^{(t)}}{\sum_{u \in \mathcal{S}^{(t)}} p_u} \tag{2}$$

The above procedure will repeat until the algorithm converges.

## 3.2 Limitation of current compression methods

Communication is one of the main bottlenecks in FedRec systems. Model transmission from server to devices can be a serious constraint for both servers and clients. For example, when stragglers with limited network connections exist, the central server must decide whether to wait for them to finish or perform the aggregation step with only available participants. Conversely, sending the model updates back to the server can be challenging, as uplink is typically much slower than downlink. The download and upload bandwidths in a real cross-device FL system are estimated at 0.75MB/s and 0.25MB/s, respectively, by Wang et al. [23]. Although diverse optimization techniques exist to enhance communication efficiency, such methods may not preserve privacy. Moreover, tackling privacy and communication efficiency as separate concerns can result in suboptimal solutions.

*Top-K compression.* The process of allocating memory for copying the gradient (which can grow to a large size, often in the millions) and then sorting this copied data to identify the top-K

threshold during each iteration is costly enough that it negates any potential enhancements in overall training time when applied to real-world systems. As a result, employing these gradient compression methods in their simplest form does not yield the expected improvements in training efficiency. As observed in Gupta et al. [7], employing the Top-K compression for training large-scale recommendation models takes 11% more time than the baseline with no compression.

*SVD compression.* After obtaining factorization results $U_u$ and $V_u$, the aggregation step requires performing decompression and computing $\sum_{u \in \mathcal{S}} \frac{N_u}{N} U_u V_u$ and this sum is not necessarily low-rank so there is no readily reducing cost in the downlink communication without additional compression-decompression step. The need to perform matrix multiplication makes this method incompatible with HE. Moreover, performing SVD decomposition on an encrypted matrix by known schemes remains an open problem.

# 4 PROPOSED METHOD

## 4.1 Motivation

Our method is motivated by analyzing the optimization process at each user's local device. We consider an effective federated matrix factorization (FedMF) as the backbone model. This model represents each item and user by a vector with the size of $d$ denoted $\mathbf{q}_i$ and $\mathbf{p}_u$ respectively. And the predicted ratings $r_{ui}$ are given by $\hat{r}_{ui} = \mathbf{q}_i^\top \mathbf{p}_u$. Then the user-wise local parameter $\theta_u$ consists of the user $u$'s embedding $\mathbf{p}_u$ and the item embedding matrix $Q$, where $\mathbf{q}_i$ is the $i$th column of $Q$. The loss function $\mathcal{L}_u$ at user $u$'s device is given in the following.

$$\mathcal{L}_u(\mathbf{p}_u, Q) = \sum_{(i, r_{ui}) \in O_u} \ell\left(r_{ui}, (Q^\top \mathbf{p}_u)_i\right) + \lambda \|\mathbf{p}_u\|_2 + \lambda \|Q\|_2 \tag{}$$

Let $\eta$ be the learning rate, the update on the user embedding $\mathbf{p}_u$ at each local optimization step is given by:

$$\mathbf{p}_u^{(t+1)} = \mathbf{p}_u^{(t)}(1 - \eta\lambda) + \eta Q^{(t)\top}(\mathbf{r} - \hat{\mathbf{r}}^{(t)}). \tag{3}$$

Let $\mathbf{m} \in \mathbb{R}^N$ be a binary vector where $\mathbf{m}_i = 1$ if $i \in O_u$, then the item embedding's $\mathbf{q}_i$ is

$$Q^{(t+1)} = Q^{(t)} - \eta(\lambda Q^{(t)} - (\mathbf{m} * (\mathbf{r}_u - \hat{\mathbf{r}}_u)) \mathbf{p}_u^{(t)\top}) \tag{4}$$

The update that is sent to the central server has the following formula,

$$\Delta_Q^{(t)} = Q^{(t+1)} - Q^{(t)} = \eta \left[ (\mathbf{m} * (\mathbf{r}_u - \hat{\mathbf{r}}_u)) \mathbf{p}_u^{(t)\top} - \lambda Q^{(t)} \right] \tag{5}$$

As we can see from equation 4, since each client only stores the presentation of only one user $\mathbf{p}_u$, the update on the item embedding matrix on each user's device at each local step are just sum of a rank-1 matrix and a regularization component. Given that $\lambda$ is typically small, the low-rank component contributes most to the update $\Delta_Q^{(t)}$. And if the direction of $\mathbf{p}_u$ does not change much during the local optimization phase, the update $\Delta_Q^{(t)}$ can stay low-rank. From this observation, we first assume that the update of the item embedding matrix in training FedRec systems $\Delta_Q^{(t)}$ can be well approximated by a low-rank matrix. We empirically verify this assumption by monitoring the effective rank of $\Delta_Q^{(t)}$ at each training round for

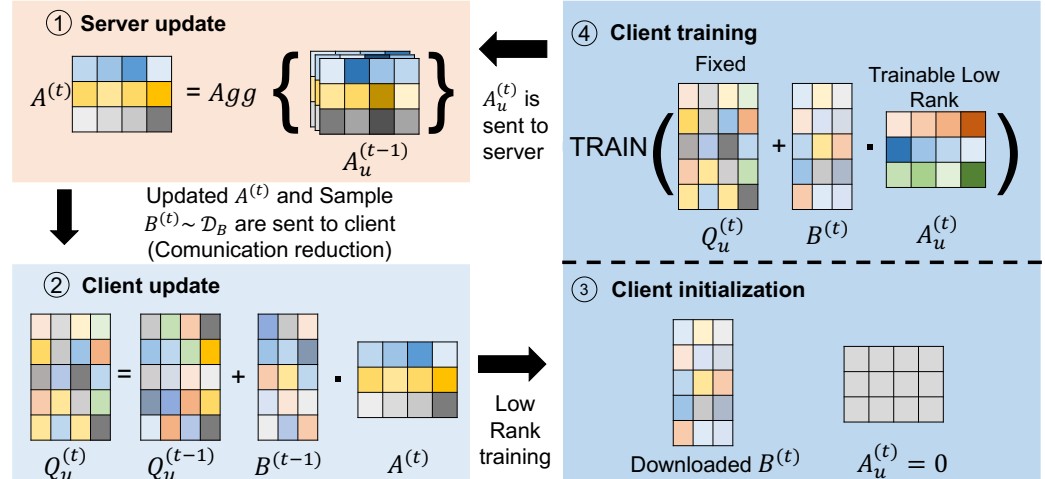

**Figure 1: Illustration of CoLR at training round $t$. At first, the server conducts aggregation over the local model $A_u^{(t-1,\tau_u)}$ to obtain the global model update $A^{(t)}$. Subsequently, $A^{(t)}$ are transmitted to the clients. The client will update their $Q_u^{(t)}$ using this $A^{(t)}$, then initilizes a new matrix $A_u^{(t,0)}$ and download the matrix $B^{(t)}$ which is sampled at the server and shared between clients. Finally, the client carries out local training and then sends the local model update $A_u^{(t,\tau_u)}$ to the server for the next training round.**

different datasets. The result is plotted in figure 2 where we plot the mean and standard deviation averaged over a set of participants in each round.

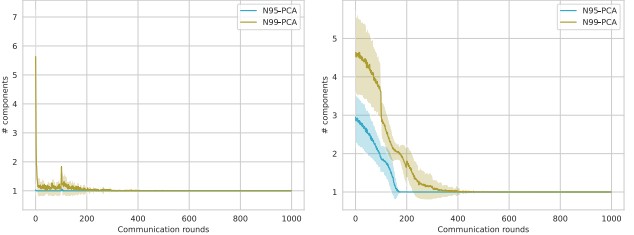

**Figure 2: PCA components progression. The figures show the number of components that account for 99% (N99-PCA in green) and 90% (N95-PCA in blue) explained variance of all transfer item embedding matrix across communication rounds on MovieLens-1M (left) and Pinterest (right) datasets.**

This analysis suggests that restricting the update to be low-rank can get away with aggressive communication reduction without sacrificing much performance. In the next section, we will propose an efficient communication framework based on this motivation. Since most of the transferred parameters in recommendation models are from the item embedding layers, we will focus on applying the proposed method for embedding layers in this work. Note that our framework can still be applied to different types of layers that are commonly used in recommendation models, such as fully connected layers, convolution layers, and self-attention layers.

## 4.2 Low-rank Structure

We propose explicitly enforcing a low-rank structure on the local update of the item embedding matrix $Q$. In particular, the local update $(\Delta_Q)_u^{(t)}$ is parameterized by a matrix product

$$(\Delta_Q)_u^{(t)} = B_u^{(t)} A_u^{(t)}$$

where $B_u^{(t)} \in \mathbb{R}^{d \times r}$ and $A_u^{(t)} \in \mathbb{R}^{r \times N}$. Given this parameterization, the embedding $\mathbf{q}_i$ of an item with index $i$ is given by

$$\mathbf{q}_i^{(t)} = \left( Q^{(t)} + B_u^{(t)} A_u^{(t)} \right) \mathbf{e}_i$$

where $\mathbf{e}_i \in \mathbb{R}^N$ is a one-hot vector whose value at $i$-th is 1. This approach effectively saves a factor of $\frac{N \times d}{N \times r + d \times r}$ in communication since clients only need to send the much smaller matrices $A_u$ and $B_u$ to the central server.

## 4.3 Correlated Low-rank Structure Update

Even though enforcing a low-rank structure on the update can greatly reduce the uplink communication size, doing aggregation and performing privacy-preserving is not trivial and faces the following three challenges: (1) the server needs to multiply out all the pairs $A_u^{(t)}$ and $B_u^{(t)}$ before performing the aggregation step; (2) the sum of low-rank solutions would typically leads to a larger rank update so there is no reducing footprint in the downlink communication; (3) secure aggregation method such as HE cannot directly apply to $A_u^{(t)}$ and $B_u^{(t)}$ since it will require to perform the multiplication between two encrypted matrices, which is much more costly than simple additive operation.

To reduce the downlink communication cost, we observe that if either $A_u^{(t)}$ or $B_u^{(t)}$ is identical between users and is fixed during the local training process, then the result of the aggregation step can be

represented by a low-rank matrix with the following formulation.

$$\Delta_Q^{(t)} = B^{(t)} \left( \sum_{u \in S} A_u^{(t)} \right).$$

Notice that this aggregation is also compatible with HE since it only requires additive operations on a set of $A_u^{(t)}$ and clients can decrypt this result and then compute the global update $\Delta_Q^{(t)}$ at their local device.

Based on the above observation, we propose the Correlated Low-rank Structure Update (CoLR) framework. In this framework, the server randomly initializes a matrix $B^{(t)}$ at the beginning of each training round and shares it among all participants. Participants then set $B_u^{(t)} = B^{(t)}$ and freeze this matrix during the local training phase and only optimize for $A_u^{(t)}$. The framework is presented in Algorithm 1 and illustrated in Figure 1. Note that the communication cost can be further reduced by sending only the random seed of the matrix $B^{(t)}$.

---

**Algorithm 1:** Correlated Low-rank Structure Update Matrix Factorization

**Input:** Initial model $Q^{(0)}$; update rank $r$, a distribution $\mathcal{D}_B$ for initializing $B$; CLIENTOPT, SERVEROPT with learning rates $\eta, \eta_s$;

1 **for** $t \in \{0, 1, 2, \ldots, T\}$ **do**
2    Sample a subset $\mathcal{S}^{(t)}$ of clients
3    Sample $B^{(t)} \sim \mathcal{D}_B$
4    **for** client $u \in \mathcal{S}^{(t)}$ **in parallel do**
5      **if** $t > 0$ **then**
6        Download $A^{(t)}$
7        Merge $Q_u^{(t,0)} = Q^{(t-1)} + B^{(t-1)} A^{(t)}$
8      **end**
9      Initialize $Q_u^{(t,0)} = Q^{(t)}$
10     Download $B^{(t)}$ and Initialize $A_u^{(t,0)} = \mathbf{0}$
11     Set trainable parameters $\theta_u^{(t,0)} = \{A_u^{(t,0)}, \mathbf{p}_u^{(t,0)}\}$
12     **for** $k = 0, \ldots, \tau_u - 1$ **do**
13       Compute local stochastic gradient $\nabla \mathcal{L}_u(\theta_u^{(t,k)})$
14       Perform local update $\theta_u^{(t,k+1)} =$
        CLIENTOPT $\left( \theta_u^{(t,k)}, \nabla_{\theta_u} \mathcal{L}_u(\theta_u^{(t,k)}), \eta \right)$
15     **end**
16     $\mathbf{p}_u^{(t+1)} = \mathbf{p}_u^{(t,\tau_u)}$
17     Upload $\{A_u^{(t,\tau_u)}\}$ to the central server
18    **end**
19    Aggregate local changes
$$A^{(t+1)} = \sum_{u \in \mathcal{S}^{(t)}} \frac{N_u}{N} A_u^{(t,\tau_u)};$$
20 **end**

---

*Differences w.r.t. SVD compression.* We compare our method with SVD since it also uses a low-rank structure. The difference is that in CoLR, participants directly optimize these models on the low-rank parameterization, while SVD only compresses the result from the local training step.

## 4.4 Subsampling Correlated Low-rank Structure update (SCoLR)

While CoLR offers its merits, there is a potential drawback to consider - it may impact recommendation performance as it confines the global update within a randomly generated low-rank subspace. In the following section, we introduce a modification to this base algorithm, considering a practical reality: downlink bandwidth often surpasses uplink capacity, as observed in cross-device scenarios [23]. In these settings, edge devices establish communication with a central server using network connections that vary in quality. Practical implementations have demonstrated significant differences in network bandwidth between download and upload capabilities. We propose a variant of CoLR termed Subsampling Correlated Low-rank Structure update (SCoLR) to address this. SCoLR strategically harnesses the more abundant downlink bandwidth while maintaining communication efficiency and HE compatibility.

We denote $r_g$ as the rank of global update, which is sent from the server to participants through downlink connections, and $r_u$ as the rank of local update, which is sent from clients to the central server for aggregation through uplink connections. In practice, we can set $r_g$ to be larger than $r_u$, reflecting that downlink bandwidth is often higher than uplink. Given these rank parameters, at the start of each training round, the central server first initializes a matrix $B$ with the shape of $\mathbb{R}^{d \times r_g}$. Then, participants in that round will download this matrix to their local devices and select a subset of columns of $B$ to perform the local optimization step. In particular, we demonstrate this process through the following formulation:

$$(\Delta_Q)_u^{(t)} = B S_u A_u \qquad (6)$$

where $B$ is a matrix with the shape of $\mathbb{R}^{d \times r_g}$, $S_u$ is a binary matrix with the shape of $\mathbb{R}^{r_g \times r_u}$ and $A_u$ is a matrix with the shape of $\mathbb{R}^{r_u \times N}$. Specifically, $S_u$ is a binary matrix with $r_u$ rows and $r_g$ columns, where each row has exactly one non-zero element. The non-zero element in the $i$-th row is at the $j$-th column, where $j$ is the $i$-th element of a randomly shuffled array of integers from 1 to $r_g$. The detail is presented in Algorithm 2.

Moreover, in scenarios where clients have diverse computational resources, each can choose a unique local rank, denoted as $r_u$, aligning with their specific computational capacities and individual preferences throughout the training process. Importantly, sharing the matrix $S_u$ does not divulge sensitive user information. Multiplying this matrix with $S_u$ is essentially a row reordering operation on the matrix $A_u$. As a result, we can effectively perform additive HE between pairs of rows from $A_{u_1}$ and $A_{u_2}$. This approach ensures privacy while accommodating varying computational capabilities among clients.

## 5 EXPERIMENTS

### 5.1 Experimental Setup

*Datasets.* We experiment with two publicly available datasets, which are MovieLens-1M [8] and Pinterest [6]. Table 1 summarizes

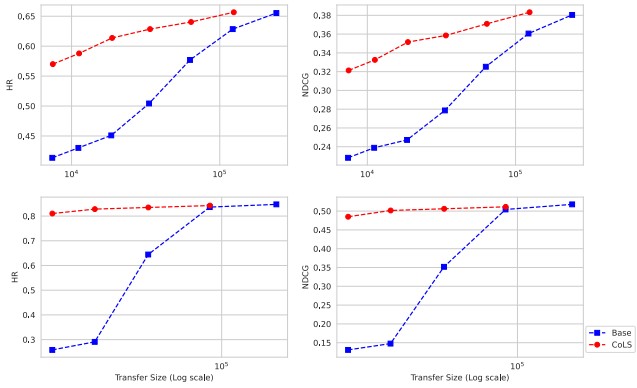

**Figure 3: HR and NDCG on MovieLens-1M dataset (Top) and Pinterest Dataset (Bottom). We plot the utilities versus the communication size and compare CoLRwith the base model with the same transfer size. The x-axis is plotted using a log scale.**

**Table 1: Statistics of the datasets used in evaluation.**

| Datasets | # Users | # Items | # Ratings | Data Density |
|---|---|---|---|---|
| MovieLens-1M [8] | 6,040 | 3,706 | 1,000,209 | 4.47% |
| Pinterest [6] | 55187 | 9916 | 1,500,809 | 0.27% |

the statistics of our datasets. We follow common practice in recommendation systems for preprocessing by retaining users with at least 20 interactions and converting numerical ratings into implicit feedback [2, 9].

*Evaluation Protocols.* We employ the standard leave-one-out evaluation to set up our test set [9]. For each user, we use all their interactions for training while holding out their last interaction for testing. During the testing phase, we randomly sampled 99 non-interacted items for each user and ranked the test item amongst these sampled items.

To evaluate the performance and verify the effectiveness of our model, we utilize two commonly used evaluation metrics, i.e., Hit Ratio (HR) and Normalized Discounted Cumulative Gain (NDCG), which are widely adopted for item ranking tasks. The above two metrics are usually truncated at a particular rank level (e.g. the first $k$ ranked items) to emphasize the importance of the first retrieved items. Intuitively, the HR metric measures whether the test item is present on the top-$k$ ranked list, and the NDCG metric measures the ranking quality, which comprehensively considers both the positions of ratings and the ranking precision.

*Models and Optimization.* For the base models, we adopt Matrix Factorization with the FedAvg learning algorithm, also used in Chai et al. [5]. In our experiments, the dimension of user and item embedding $d$ is set to 64 for the MovieLens-1M dataset and 16 for the Pinterest dataset. This is based on our observation that increasing the embedding size on the Pinterest dataset leads to overfitting and decreased performance on the test set. The result is also consistent

with [9]. We use the simple SGD optimizer for local training at edge devices.

*Federated settings.* In each round, we sample $M$ clients uniformly randomly, without replacement in a given round and across rounds. Instead of performing $\tau_i$ steps of ClientOpt, we perform $E$ epochs of training over each client's dataset. This is because, in practical settings, clients have heterogeneous datasets of varying sizes. Thus, specifying a fixed number of steps can cause some clients to repeatedly train on the same examples, while certain clients only see a small fraction of their data.

*Baselines.* We compare our framework with two compression methods: SVD and TopK compression. The first method is based on singular value decomposition, which returns a compressed update with a low-rank structure. The second method is based on sparsification, which represents updates as sparse matrices to reduce the transfer size.

*Hyper-parameter settings.* To determine hyper-parameters, we create a validation set from the training set by extracting the second last interaction of each user and tuning hyper-parameters on it. We tested the batch size of [32, 64, 128, 256], the learning rate of [0.5, 0.1, 0.05, 0.01], and weight decay in [$5e-4, 1e-4$]. For each dataset, we set the number of clients participating in each round to be equal to 1%. The number of aggregation epochs is set at 1000 for MovieLens-1M and 2000 for Pinterest as the training process is converged at these epochs.

*Machine.* The experiments were conducted on a machine equipped with an Intel(R) Xeon(R) W-1250 CPU @ 3.30GHz and a Quadro RTX 4000 GPU.

## 5.2 Experimental Results

**(1) CoLR can achieve comparable performances with the base models.** Given our primary focus is on recommendation performance within communication-limited environments, we commence our investigation by comparing the recommendation performance between CoLR and the base model given the same communication budget. For our base model, we implement FedMF with the FedAvg learning algorithm. For the ML-1M dataset, We adjust the dimensions of user and item embeddings across the set [1, 2, 4, 8, 16, 32, 64] for FedMF while fixing the embedding size of CoLR to 64, with different rank settings within [1, 2, 4, 8, 16, 32]. Similarly, for Pinterest, the embedding range for FedMF is [1, 2, 4, 8, 16], while CoLR has an embedding size of 16 and ranks in the range of [1, 2, 4, 8]. Our settings lead to approximately equivalent transfer sizes for both methods in each dataset.

In Figure 3, we present the HR and NDCG metrics across a range of different transfer sizes. With equal transfer budget on the Pinterest dataset, CoLR consistently outperforms their counterparts. To illustrate, even with an update size equates to 6.25% of the largest model, CoLR achieves a notable performance (81.03% HR and 48.50% NDCG) compared to the base model (84.74% HR and 51.79% NDCG) while attaining a much larger reduction in terms of communication cost (16x). In contrast, the FedMF models with corresponding embedding sizes achieve much lower accuracies. On the MovieLens-1M dataset, we also observe a similar pattern

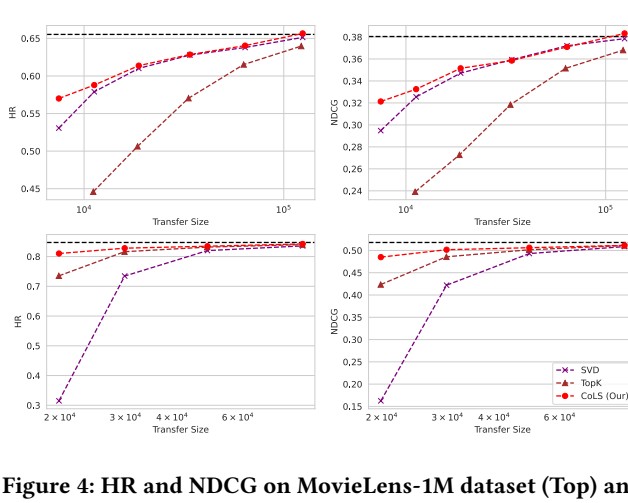

**Figure 4: HR and NDCG on MovieLens-1M dataset (Top) and Pinterest Dataset (Bottom). We plot the utilities versus the communication size and compare CoLRwith the base model with the same transfer size. The x-axis is plotted using a log scale. The dashed black line is the base model's performance with different embedding sizes. For fair comparisons, we set the compression ratio of each method to be equal.**

where CoLR consistently demonstrates lower training losses when compared to their counterparts.

The result from this experiment highlights that CoLR can achieve competitive performance when compared to the base model, FedMF while greatly reducing the cost of communication.

**Table 2: Communication times and Training times for MovieLens-1M dataset.**

| Method | Communication time (minutes) | Computation time (minutes) | Total Training Time (minutes) |
|---|---|---|---|
| MF-64 | 80.43 | 169.07 | 249.50 |
| CoLR@1 | 1.26 | 169.18 | 170.43 |
| CoLR@2 | 2.51 | 169.21 | 171.72 |
| CoLR@4 | 5.03 | 169.27 | 174.30 |
| CoLR@8 | 10.05 | 169.29 | 179.34 |
| CoLR@16 | 20.11 | 169.30 | 189.41 |
| CoLR@32 | 40.21 | 169.38 | 209.60 |
| SVD@1 | 1.26 | 169.49 | 170.75 |
| SVD@2 | 2.51 | 169.50 | 172.02 |
| SVD@4 | 5.03 | 169.53 | 174.55 |
| SVD@8 | 10.05 | 169.59 | 179.65 |
| SVD@16 | 20.11 | 169.64 | 189.74 |
| SVD@32 | 40.21 | 169.60 | 209.82 |
| TopK@1 | 2.51 | 169.76 | 172.28 |
| TopK@2 | 5.03 | 169.79 | 174.81 |
| TopK@4 | 10.05 | 169.82 | 179.87 |
| TopK@8 | 20.11 | 169.92 | 190.03 |
| TopK@16 | 40.21 | 170.14 | 210.35 |

**(2) Comparison between CoLR and other compression-based methods.** We run the above experiment with two compression methods, SVD and TopK, with the compression ratio about the same as CoLR. For a fair comparison, we compress both the upload and download messages with the same compression ratio. The result in terms of transfer size is presented in Figure 4. In the case with the same communication budget, CoLR achieves better performance across the range of communication budgets.

In the previous results, the evaluation of techniques focuses on the overall number of transmitted bits. Although this serves as a broad indicator, it fails to consider the time consumed by encoding/decoding processes and the fixed network latency within the system. When these time delays significantly exceed the per-bit communication time, employing compression techniques may offer limited or minimal benefits. In the following, we do an analysis to understand the effects of using CoLR and compression methods in training FedRec models.

We follow the model from [23] to estimate the communication efficiency of deploying methods to real-world systems. The execution time per round when deploying an optimization algorithm in a cross-device FL system is estimated as follows,

$$T_{\text{round}}(\mathcal{A}) = T_{\text{comm}}(\mathcal{A}) + T_{\text{comp}}(\mathcal{A}),$$

$$T_{\text{comm}}(\mathcal{A}) = \frac{S_{\text{down}}(\mathcal{A})}{B_{\text{down}}} + \frac{S_{\text{up}}(\mathcal{A})}{B_{\text{up}}}$$

$$T_{\text{comp}}(\mathcal{A}) = \max_{j \in \mathcal{D}_{\text{round}}} T_{\text{client}}^{j} + T_{\text{server}}(\mathcal{A}),$$

$$T_{\text{client}}^{j}(\mathcal{A}) = R_{\text{comp}} T_{\text{sim}}^{j}(\mathcal{A}) + C_{\text{comp}}$$

where client download size $S_{\text{down}}(\mathcal{A})$, upload size $S_{\text{up}}(\mathcal{A})$, server computation time $T_{\text{server}}$, and client computation time $T_{\text{client}}^{j}$ depend on model and algorithm $\mathcal{A}$. Simulation time $T_{\text{server}}$ and $T_{\text{client}}^{j}$ can be estimated from FL simulation in our machine. We get the estimation of parameters $(B_{\text{down}}, B_{\text{up}}), R_{\text{comp}}, C_{\text{comp}}$ from Wang et al. [23].

$$B_{\text{down}} \sim 0.75\text{MB/secs}, B_{\text{up}} \sim 0.25\text{MB/secs},$$
$$R_{\text{comp}} \sim 7, \text{ and } C_{\text{comp}} \sim 10 \text{ secs}.$$

Table 2 presents our estimation in terms of communication times and computation time. Notice that CoLR adds smaller overheads to the computation time while still greatly reducing the communication cost.

**(3) CoLR is compatible with HE.** In our experiment, we conduct tests using two different compression methods: CoLR and TopK. These tests are carried out under identical configurations, encompassing local updates and the number of clients involved in training rounds. The current setup entails the utilization of the CKKS Cryptosystem for our CoLR method, while the TopK method employs the Paillier cryptosystem for encryption, decryption, and aggregation in place of the TopK vector.

Currently, multiple open-source HE libraries are available, such as OpenFHE [1], TenSeal [3], and Microsoft SEAL [20]. We chose OpenFHE for its renowned speed and compatibility with essential operations in our federated learning framework. Regarding the runtime aspect, our CoLR compression method leverages the inherent efficiency of the CKKS cryptosystem, which can execute operations

**Table 3: Overheads, and Communication ratios for MovieLens-1M dataset; Comm Ratio is calculated by file sizes of Ciphertext over file sizes of Plaintext.**

| EC method | Client overheads (s) | Server overheads (s) | Ciphertext | Plaintext | Comm Ratio |
|---|---|---|---|---|---|
| FedMF | 0.93 | 2.39 | 24,587 KB | 927 KB | 26.52 |
| FedMF w/ TopK@1/64 | 88.2 | 88.06 | 3,028 KB | 29 KB | 103.09 |
| FedMF w/ TopK@2/64 | 182.02 | 185.59 | 6,056 KB | 58 KB | 103.83 |
| FedMF w/ TopK@4/64 | 353.25 | 364.67 | 12,112 KB | 116 KB | 104.20 |
| FedMF w/ TopK@8/64 | 723.45 | 750.98 | 24,225 KB | 232 KB | 104.40 |
| FedMF w/ TopK@16/64 | 1449.9 | 1,483.91 | 48,448 KB | 464 KB | 104.49 |
| FedMF w/ CoLR@1 | 0.07 | 0.24 | 3,073 KB | 15 KB | 206.31 |
| FedMF w/ CoLR@2 | 0.07 | 0.25 | 3,073 KB | 29 KB | 104.63 |
| FedMF w/ CoLR@4 | 0.07 | 0.25 | 3,073 KB | 58 KB | 52.69 |
| FedMF w/ CoLR@8 | 0.08 | 0.25 | 3,073 KB | 116 KB | 26.44 |
| FedMF w/ CoLR@16 | 0.15 | 0.51 | 6,147 KB | 232 KB | 26.49 |
| FedMF w/ CoLR@32 | 0.30 | 1.03 | 12,293 KB | 464 KB | 26.51 |

on multiple values as a vector. In our approach, for a flattened vector of size $n$, both clients and the server need to perform operations on at most $\lceil \frac{n}{8096} \rceil$ blocks, which incurs minimal computational time. On the other hand, the TopK method necessitates operations to be executed on every value within the TopK vector, leading us to opt for the Paillier cryptosystem as it is partially homomorphic and can accommodate the requirements of both our schemes and the TopK method. In our experimental setup, when the value of $k$ doubles (i.e., doubling the TopK vector's size), the operation time for both client-side and server-side operations also doubles, as it mandates operations on each value within the vector. Throughout the experiment, our CoLR consistently outperforms the TopK method across various $k$ values, exhibiting lower time overheads on both the client and server sides.

When comparing ciphertext to plaintext sizes, the TopK compression method with Paillier encryption demands encryption for each value within the TopK vector. Consequently, whenever the size of the TopK vector doubles, the ciphertext size also doubles. In contrast, as previously explained, our scheme produced at most $\lceil \frac{n}{8096} \rceil$ blocks of ciphertext, with the ciphertext size not doubling each time $k$ doubles. This phenomenon illustrates why, in several cases, the ciphertext size remains consistent even as the plaintext size increases. With higher $k$ values aimed at achieving greater precision, our scheme demonstrates smaller ciphertext sizes, offering a reduction in bandwidth consumption.

## 5.3 Dynamic local rank

In this section, we evaluate our proposed method SCoLR, where we explore the scenario where each client can dynamically select a random $r_u$ value during each training round $t$. This scenario reflects real-world federated learning, where clients often showcase differences in computing capabilities and communication capacities due to hardware discrepancies, as exemplified in [12, 13]. It becomes inefficient to impose a uniform training model on all clients within this heterogeneous context, as some devices may not be able to harness their computational resources fully.

For this experiment, we set the global rank $r$ in range $\{4, 8, 16, 32\}$ and uniformly sample the local rank $r_u$ such that $1 \leq r_u \leq r$. It's

crucial to emphasize that $r_u$ is independently sampled for each user and may differ from one round to the next. This configuration mirrors a practical scenario where the available resources of a specific user may undergo substantial variations at different time points during the training phase. We present the result on the MovieLens-1M dataset in Table 4. This result demonstrates that SCoLR is effective under the device heterogeneity setting.

**Table 4: HR, NDCG of SCoLR algorithm on the MovieLens-1M dataset under computation/device heterogeneity settings.**

| Global low-rank | Local-rank | HR | NDCG |
|---|---|---|---|
| 32 | $1 - 32$ | 64.59 | 37.56 |
| 16 | $1 - 16$ | 60.25 | 34.33 |
| 8 | $1 - 8$ | 52.75 | 28.41 |
| 4 | $1 - 4$ | 45.99 | 23.70 |

## 6 CONCLUSION

In this work, we propose Correlated Low-rank Structure update (CoLR), a framework that enhances communication efficiency and privacy preservation in FedRec by leveraging the inherent low-rank structure in updating transfers, our method reduces communication overheads. CoLR also benefits from the CKKS cryptosystem, which allows the implementation of a secured aggregation strategy within FedRec. With minimal computational overheads and bandwidth-heterogeneity awareness, it offers a flexible and efficient means to address the challenges of federated learning. For future research, we see several exciting directions. First, our framework still involves a central server, we would like to test how our methods can be effectively adapted to a fully decentralized, peer-2-peer communication setting. Secondly, investigating methods to handle dynamic network conditions and straggler mitigation in real-world settings will be crucial. Lastly, expanding our approach to accommodate more advanced secure aggregation techniques for reduced server-side computational costs and extending its compatibility with various encryption protocols can further enhance its utility in privacy-sensitive applications.

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

## A AN ANALYSIS ON THE INITIALIZATION OF THE MATRIX B

If each client performs only one GD step locally then $B$ can be seen as the projection matrix and $B\mathbf{a}_i$ is the projection of the update of item $i$ on the subspace spanned by columns of $B$. We denoite the error of the update on each item embedding $i$ by $\epsilon_i$ which has the following formulation:

$$\epsilon_i = \mathbb{E}_B \left[ \left\| \bar{\Delta}_Q - \frac{1}{|S|} \left( \sum_{u \in S} B_u \mathbf{a}_u \right) \right\|_2^2 \right]. \tag{7}$$

We analyze the effect of different initialization of $B$ on this error. First, we state the proposition A.1 which gives an upper bound on the error $\epsilon_i$.

PROPOSITION A.1 (UPPER BOUND THE ERROR). *If $B_u$ is independently generated between users and are chosen from a distribution $\mathcal{B}$ that satisfies:*

(1) *Bounded operator norm:* $\mathbb{E}\left[\|B\|^2\right] \le L_{\mathcal{B}}$
(2) *Bounded bias:* $\|\mathbb{E}B_u B_u^\top \bar{\mathbf{p}}_u - \bar{\mathbf{p}}_u\|_2 \le \sqrt{\delta_{\mathcal{B}}}$

*Then,*

$$\epsilon_i = \mathbb{E}_B \left[ \left\| \bar{\Delta}_Q - \frac{1}{|S|} \left( \sum_{u \in S} B_u \mathbf{a}_u \right) \right\|_2^2 \right] \tag{8}$$

$$\le \frac{1}{|S|} C_{\mathbf{p}}^2 \delta_{\mathcal{B}} + \frac{1}{|S|} \max_{u \in S} \alpha_u \|\mathbf{p}_u\|_2^2 \left( L_B^2 + 1 \right). \tag{9}$$

PROOF. Assume $B_u$ is independently generated between users, we have

$$\epsilon_i = \frac{1}{|S|^2} \mathbb{E}_B \left[ \left\| \sum_{u \in S} (r_{ui} - \hat{r}_{ui}) \left(B_u B_u^\top \mathbf{p}_u - \mathbf{p}_u\right) \right\|_2^2 \right]$$

$$= \frac{1}{|S|^2} \mathbb{E}_B \left[ \left\| \sum_{u \in S} \alpha_u \left(B_u B_u^\top \mathbf{p}_u - \mathbf{p}_u\right) \right\|_2^2 \right]$$

$$= \frac{1}{|S|^2} \sum_{u_1 \in S} \sum_{u_2 \neq u_1} \alpha_{u_1} \alpha_{u_2} \mathbb{E}_B \left\langle B_{u_1} B_{u_1}^\top \mathbf{p}_{u_1} - \mathbf{p}_{u_1}, B_{u_2} B_{u_2}^\top \mathbf{p}_{u_2} - \mathbf{p}_{u_2} \right\rangle$$

$$+ \frac{1}{|S|^2} \sum_{u \in S} \alpha_u^2 \mathbb{E}_B \left[ \left\| B_u B_u^\top \mathbf{p}_u - \mathbf{p}_u \right\|_2^2 \right]$$

If $B_u$ are independently chosen from a distribution $\mathcal{B}$ that satisfies:

(1) Bounded operator norm: $\mathbb{E}\left[\|B\|^2\right] \le L_{\mathcal{B}}$
(2) Bounded bias: $\|\mathbb{E}B_u B_u^\top \bar{\mathbf{p}}_u - \bar{\mathbf{p}}_u\|_2 \le \sqrt{\delta_{\mathcal{B}}}$

We have

$$\mathbb{E}_B \left\langle B_{u_1} B_{u_1}^\top \mathbf{p}_{u_1} - \mathbf{p}_{u_1}, B_{u_2} B_{u_2}^\top \mathbf{p}_{u_2} - \mathbf{p}_{u_2} \right\rangle$$

$$= \left\| \mathbf{p}_{u_1} \right\| \left\| \mathbf{p}_{u_2} \right\| \mathbb{E}_B \left\langle B_{u_1} B_{u_1}^\top \bar{\mathbf{p}}_{u_1} - \bar{\mathbf{p}}_{u_1}, B_{u_2} B_{u_2}^\top \bar{\mathbf{p}}_{u_2} - \bar{\mathbf{p}}_{u_2} \right\rangle$$

$$= \left\| \mathbf{p}_{u_1} \right\| \left\| \mathbf{p}_{u_2} \right\| \left\langle \mathbb{E}B_{u_1} B_{u_1}^\top \bar{\mathbf{p}}_{u_1} - \bar{\mathbf{p}}_{u_1}, \mathbb{E}B_{u_2} B_{u_2}^\top \bar{\mathbf{p}}_{u_2} - \bar{\mathbf{p}}_{u_2} \right\rangle \tag{10}$$

$$\le \left\| \mathbf{p}_{u_1} \right\| \left\| \mathbf{p}_{u_2} \right\| \|\mathbb{E}B_{u_1} B_{u_1}^\top \bar{\mathbf{p}}_{u_1} - \bar{\mathbf{p}}_{u_1}\|_2 \|\mathbb{E}B_{u_2} B_{u_2}^\top \bar{\mathbf{p}}_{u_2} - \bar{\mathbf{p}}_{u_2}\|_2$$

$$\le C_{\mathbf{p}}^2 \delta_{\mathcal{B}} \tag{11}$$

where (11) follows since $B_u$ are independently sampled between users. The second term is

$$\frac{1}{|S|^2} \sum_{u \in S} \alpha_u^2 \mathbb{E}_B \left[ \left\| B_u B_u^\top \mathbf{p}_u - \mathbf{p}_u \right\|_2^2 \right]$$

$$= \frac{1}{|S|^2} \sum_{u \in S} \alpha_u^2 \|\mathbf{p}_u\|_2^2 \mathbb{E}_B \left[ \left\| B_u B_u^\top \bar{\mathbf{p}}_u - \bar{\mathbf{p}}_u \right\|_2^2 \right]$$

$$= \frac{1}{|S|^2} \sum_{u \in S} \alpha_u \|\mathbf{p}_u\|_2^2 \mathbb{E}_B \left[ \left\| B_u B_u^\top \bar{\mathbf{p}}_u \right\|_2^2 + \|\bar{\mathbf{p}}_u\|_2^2 - 2 \bar{\mathbf{p}}_u^\top B_u B_u^\top \bar{\mathbf{p}}_u \right]$$

$$= \frac{1}{|S|^2} \sum_{u \in S} \alpha_u \|\mathbf{p}_u\|_2^2 \mathbb{E}_B \left[ \left\| B_u B_u^\top \bar{\mathbf{p}} \right\|_2^2 + 1 - 2 \left\| B_u^\top \bar{\mathbf{p}} \right\|_2^2 \right]$$

$$\le \frac{1}{|S|} \max_{u \in S} \alpha_u \|\mathbf{p}_u\|_2^2 \left( \mathbb{E}_B \left[ \left\| B_u B_u^\top \right\|_2^2 \right] + 1 \right)$$

$$\le \frac{1}{|S|} \max_{u \in S} \alpha_u \|\mathbf{p}_u\|_2^2 \left( L_B^2 + 1 \right)$$

□

Next, we bound the bias and the operator norm of $B_u$ if it is sampled from a Gaussian distribution in the lemma A.2.

LEMMA A.2 (GAUSSIAN INITIALIZATION). *Let $r < d$. Consider $B \in \mathbb{R}^{d \times r}$ be sampled from the Gaussian distribution where $B$ has i.i.d. $\mathcal{N}(0, 1/k)$ entries and a fixed unit vector $\mathbf{v} \in \mathbf{R}^d$. Then*

(1) *Bounded operator norm:*

$$\mathbb{E}\|B\|^2 \le \frac{d}{r} \left( 1 + O\left( \sqrt{\frac{r}{d}} \right) \right)$$

(2) *Unbias: for every unit vector $\mathbf{v} \in \mathbb{R}^d$*

$$\left\| \mathbb{E}BB^\top \mathbf{v} - \mathbf{v} \right\| = 0$$

PROOF. Let $B' = PB$ where $P \in \mathbb{R}^{d \times d}$ is the rotation matrix such that $P\mathbf{v} = \mathbf{e}_1$. Due to the rotational symmetry of the normal distribution, $B'$ is a random matrix with i.i.d. $\mathcal{N}(0, 1/r)$ entries. Note that $B = P^\top B'$.

$$\mathbb{E}_B \left[ BB^\top \mathbf{v} \right] = \mathbb{E}_B \left[ P^\top PBB^\top P^\top P\mathbf{v} \right]$$

$$= P^\top \mathbb{E}_B \left[ B'B'^\top \mathbf{e}_1 \right]$$

Let $\mathbf{z} = B'B'^\top \mathbf{e}_1$. Notice that $\mathbf{z}_j = \left\langle B'^\top \mathbf{e}_j, B'^\top \mathbf{e}_1 \right\rangle$. Because $B'$ has i.i.d. $\mathcal{N}(0, 1/r)$ entries, $\mathbf{z}_1 = \|B'^\top \mathbf{e}_1\|_2^2 = \sum_{k=1}^r (B'_{1k})^2$ is $1/r$ times a Chi-square random variable with $r$ degrees of freedom. So $\mathbb{E}[\mathbf{z}_1] = \frac{1}{r} r = 1$ and $\mathbb{E}[\mathbf{z}_j] = 0 \forall j > 1$. Thus, $\mathbb{E}[\mathbf{z}] = \mathbf{e}_1$. Therefore, $\left\| \mathbb{E}_B \left[ BB^\top \mathbf{v} \right] \right\| = \left\| P^\top \mathbf{e}_1 - \mathbf{v} \right\| = 0$. □

From Proposition A.1 and Lemma A.2, we can directly get the following theorem which bould the error of restricting the local update in a low-rank subspace which is randomly sampled from a normal distribution.

THEOREM A.3. *Assume $B_u$ is independently generated between users and are chosen from the normal distribution $\mathcal{N}(0, 1/r)$. Then,*

$$\epsilon_i = \mathbb{E}_B \left[ \left\| \bar{\Delta}_Q - \frac{1}{|S|} \left( \sum_{u \in S} B_u \mathbf{a}_u \right) \right\|_2^2 \right]$$

$$\le \frac{1}{|S|} \max_{u \in S} \alpha_u \|\mathbf{p}_u\|_2^2 O\left( \frac{d}{r} \right).$$

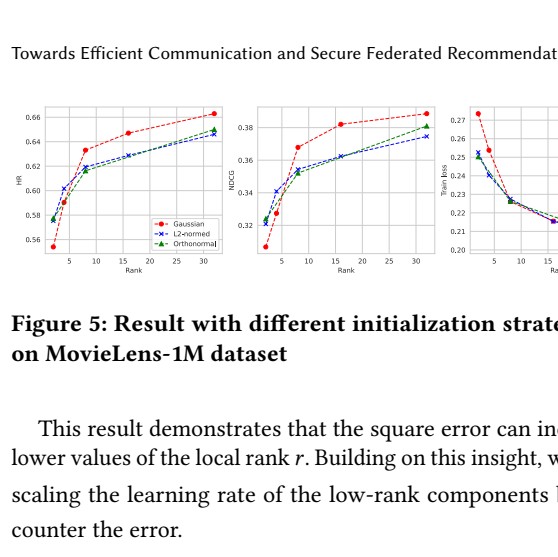

**Figure 5: Result with different initialization strategy for $B$ on MovieLens-1M dataset**

This result demonstrates that the square error can increase for lower values of the local rank $r$. Building on this insight, we suggest scaling the learning rate of the low-rank components by $\sqrt{\frac{r}{d}}$ to counter the error.

*Experementing with different initialization strategies for $B$.* The result in the main text is reported using $B$ sampled from a normal distribution $\mathcal{N}(0, 1/r)$. In this section, we experiment with $\mathcal{D}_B$ chosen from a normal distribution, distribution of orthonormal matrix, and Gaussian distribution on a unit sphere. The result is s

## B ALGORITHM DETAILS

In Section 4.4, we presented SCoLR to address the bandwidth heterogeneity problem. We provide the detail of this method in Algorithm 2.

---

**Algorithm 2:** Subsampling Correlated Low-rank Structure update (SCoLR)

---

**Input:** Initial model $Q^{(0)}$; global update rank $r_g$, local update rank $\{r_u\}$, a distribution $\mathcal{D}_B$ for initializing $B$; CLIENTOPT, SERVEROPT with learning rates $\eta, \eta_s$;

1 **for** $t \in \{0, 1, 2, \ldots, T\}$ **do**

2    Sample a subset $\mathcal{S}^{(t)}$ of clients

3    Sample $B^{(t)} \sim \mathcal{D}_B$

4    **for** client $u \in \mathcal{S}^{(t)}$ **in parallel do**

5      **if** $t > 0$ **then**

6        Download $A^{(t)}$

7        Merge $Q_u^{(t,0)} = Q^{(t-1)} + B^{(t-1)}A^{(t)}$

8      **end**

9      Initialize $Q_u^{(t,0)} = Q^{(t)}$

10      Download $B^{(t)}$, Initialize $A_u^{(t,0)} = \mathbf{0}$, and Randomly sample $S_u^{(t)}$

11      Set trainable parameters $\theta_u^{(t,0)} = \{A_u^{(t,0)}, \mathbf{p}_u^{(t,0)}\}$

12      **for** $k = 0, \ldots, \tau_u - 1$ **do**

13        Compute local stochastic gradient $\nabla\mathcal{L}_u(\theta_u^{(t,k)})$

14        Perform local update $\theta_u^{(t,k+1)} = $ CLIENTOPT $\left(\theta_u^{(t,k)}, \nabla_{\theta_u}\mathcal{L}_u(\theta_u^{(t,k)}), \eta\right)$

15      **end**

16      $\mathbf{p}_u^{(t+1)} = \mathbf{p}_u^{(t,\tau_u)}$

17      Upload $\{S_u^{(t)}, A_u^{(t,\tau_u)}\}$ to the central server

18    **end**

19    Aggregate local changes

$$A^{(t+1)} = \sum_{u \in \mathcal{S}^{(t)}} \frac{N_u}{N} S_u^{(t)} A_u^{(t,\tau_u)};$$

20 **end**

---

