# OpenReview forum: "Towards Efficient Communication and Secure Federated Recommendation System via Low-rank Training"
_ACM.org/TheWebConf/2024/Conference — TheWebConf24 Oral_

### Official Review · Reviewer_Aquc · 2023-11-21

**Novelty:** 5
**Technical Quality:** 6

**Review:**

Advantages:
1. Quality: This work provides a detailed description of the recommendation system method for federated learning, covering key elements such as model architecture, loss function, and parameter updates, enabling readers to understand the working principle and implementation steps of the method.
2. Clarity: Clear language and symbols are used in the text to enable readers to accurately understand the mathematical expressions and steps of the algorithm. The definition and interpretation of key concepts help ensure that readers can understand the author's intentions.
3. Originality: This work introduces a recommendation system method based on federated learning and proposes two communication frameworks: Correlated Low rank Structure Update (CoLR) and Subsampling Correlated Low rank Structure Update (SCoLR) to reduce communication overhead and improve privacy protection. These frameworks have a certain degree of innovation in the field of federated learning.
4. Significance: Federated learning is of great significance in privacy protection and data security, especially in applications involving personal information such as recommendation systems. The methods and communication framework proposed in this work provide new ideas and methods for addressing communication and privacy issues in federated learning.

Disadvantages:
1. The limitations of the method and the applicability of practical application scenarios were not fully discussed in this work.

**Questions:**

1、What is the performance of this method in practical applications?
2、How scalable is this framework on large-scale datasets and complex models? Is there a problem of communication overhead increasing with the size of the dataset?
3. Is the communication framework mentioned in the text applicable to all federated learning scenarios, or is it only applicable to specific types of problems or data distributions?
4、Is the method in the text feasible for practical deployment?
5. Are there any limitations to this method? In what situations may this method not be applicable or performance degradation occur?

**Reviewer Confidence:**

3: The reviewer is confident but not certain that the evaluation is correct

**Scope:**

4: The work is relevant to the Web and to the track, and is of broad interest to the community

---

### Official Review · Reviewer_Un2X · 2023-11-22

**Novelty:** 7
**Technical Quality:** 7

**Review:**

The authors propose a framework inspired by parameter-efficient fine-tuning to increase efficiency in federated learning updates. They propose a framework called Correlated Low-rank Structure update (CoLR) that uses low-rank updates to enhance communication efficiency in updating transfers.

Strengths:
The paper is overall very well written and motivated. All important concepts were introduced thoroughly and related work gives a clear relation to their approach. The work seems novel to use LoRA approaches within federated learning and test it thoroughly in their experiments. The approach seems to be a very competitive new approach in efficient federated learning.

Weaknesses:
The author’s propose to only update A and instantiate B randomly. It is not clear to me why B is instantiated randomly and why B should not be used to capture information (updated within the model).

Some nitpicks:
In the introduction the authors claim
"Based on our observation that the update transferred between clients and a central server has a low-rank structure bias, we propose Correlated Low-rank Structure update (CoLR).” At this point in the paper it is not clear why this observation holds. Maybe the authors can add some motivational details.

**Questions:**

Why is B initialized randomly?

**Reviewer Confidence:**

2: The reviewer is willing to defend the evaluation, but it is likely that the reviewer did not understand parts of the paper

**Scope:**

4: The work is relevant to the Web and to the track, and is of broad interest to the community

---

### Official Review · Reviewer_VEa9 · 2023-11-24

**Novelty:** 3
**Technical Quality:** 3

**Review:**

Strong points：

1.	The authors propose a framework, Correlated Low-rank Structure update (CoLR), designed to enhance communication efficiency and privacy protection in FedRec. By leveraging the inherent low-rank structure in updating transfers, the method effectively reduces communication overhead.

2.	The paper provides a thorough and persuasive explanation of the motivation behind the research.

3.	The experiments consider several baselines and datasets. The experimental results demonstrate a significant improvement in efficiency achieved by the proposed method.

Weak points:

1.	The paper lacks substantial innovation, as the proposed mechanism merely integrates federated learning for recommender systems with LoRA. The contribution appears to be primarily a combination of existing methodologies rather than a novel approach or advancement in the field.

2.	The title of the paper emphasizes both efficiency and security as crucial optimization directions for the proposed model. However, upon closer examination of the content, the security optimization seems limited to enabling the aggregation of Homomorphic Encryption, and this choice appears to be primarily driven by efficiency considerations. As a result, the paper lacks comprehensive exploration and experimentation regarding security aspects.

3.	In Section 4.3, the paper introduces a strategy of freezing the B matrix within the LoRA mechanism. This measure appears to be aimed at enhancing the efficiency of aggregating Homomorphic Encryption. However, this strategy lacks theoretical underpinning and experimental validation, both of which are essential for substantiating its effectiveness.

4.	The paper needs a more comprehensive review of recent relevant research.

a)	Liang F, Pan W, Ming Z. Fedrec++: Lossless federated recommendation with explicit feedback[C]//Proceedings of the AAAI conference on artificial intelligence. 2021, 35(5): 4224-4231.

b)	Liu S, Xu S, Yu W, et al. FedCT: Federated collaborative transfer for recommendation[C]//Proceedings of the 44th international ACM SIGIR conference on research and development in information retrieval. 2021: 716-725.

c)	Yuan W, Nguyen Q V H, He T, et al. Manipulating Federated Recommender Systems: Poisoning with Synthetic Users and Its Countermeasures[J]. arXiv preprint arXiv:2304.03054, 2023.

5.	There are some errors in the texts that need to be rectified.

a)	At line 121 of the paper, pre-trained parameters are mentioned, even though the paper does not incorporate a pretraining module.

b)	At line 183, in the introduction of Communication Efficient Federated Recommendation, the concluding phrase is 'However, these.'

6.	Further explanation is needed for the experimental setup and results.

a)	This paper lacks a fundamental introduction to the algorithms employed in the baseline used in this study.

b)	The baseline in this paper comprises only traditional algorithms, lacking some of the latest algorithms in the field, such as MetaMF and LightFR mentioned in the Communication Efficient Federated Recommendation section of the related work.

c)	The experimental section lacks sufficient explanations for the experimental charts, and there are also some points of confusion.

i.	In Figure 4, the black dashed line is explained as 'the base model’s performance with different embedding sizes,' but each plot in the figure contains only one black dashed line.

ii.	Table 2 is not referenced in the paper, and the results are not explained.

d)	The only experiment pertaining to security in the experimental section is found in Table 3. However, this table compares efficiency aspects, lacking experiments specifically addressing security.

e)	The experiments regarding SCoLR lack comparisons with other algorithms, rendering the results less convincing.

**Questions:**

Please refer to the weak points 1-6.

**Reviewer Confidence:**

3: The reviewer is confident but not certain that the evaluation is correct

**Scope:**

4: The work is relevant to the Web and to the track, and is of broad interest to the community

---

### Official Review · Reviewer_Zn1s · 2023-11-24

**Novelty:** 5
**Technical Quality:** 6

**Review:**

This paper studies a new approach to federated recommendation systems by using low rank training to improve on the communication efficiency. The authors propose a method called Correlated Low-rank Structure Update (CoLR), where only a small number of trainable parameters are shared between the server and clients, while most pre-trained parameters are kept frozen.

The authors start with federated matrix factorization, where they enforce a low-rank structure on the local update of the item embedding matrix by parameterizing it as a matrix product to save on communication costs. For the aggregation step to be compatible with homomorphic encryption, the authors also propose that the server randomly initializes one factor in the product in each training round, so that users only optimize for the other factor. They experiment with the MovieLens and Pinterest datasets, and they find that CoLR can achieve competitive performance compared to the base model, while greatly reducing the communication costs.

This paper is clearly written and organized, and the technical portions and experiments are described with sufficient detail.

**Questions:**

None.

**Ethics Review Description:**

None.

**Reviewer Confidence:**

2: The reviewer is willing to defend the evaluation, but it is likely that the reviewer did not understand parts of the paper

**Scope:**

4: The work is relevant to the Web and to the track, and is of broad interest to the community

---

### Decision · Program_Chairs · 2024-01-22

**Decision:**

Accept (Oral)

**Comment:**

This paper leverages the idea of parameter-efficient fine-tuning to achieve communication efficient and secure federated recommendation. Under the setting of federated matrix factorization, the proposed solution chose to share only a small number of trainable parameters among the server and clients to reduce communication cost. Competitive performance was achieved on two public recommendation benchmark datasets.

 All reviewers appreciated the writing quality of this manuscript and its encouraging empirical performance. The reviewers also pointed out the paper's contribution on the security side is limited, though it is emphasized in the title.

 Overall, the evaluation of this paper is very positive and we are happy to recommend accepting this work, but we still suggest the authors to further polish the paper's content according to the reviewers' comments and suggestions to make the content more appreciated by the WWW community.